# FINE-TUNING MLLMS WITHOUT FORGETTING IS EASIER THAN YOU THINK

## ABSTRACT

The paper demonstrate that simple adjustments of the fine-tuning recipes of multimodal large language models (MLLM) are sufficient to mitigate catastrophic forgetting. On visual question answering, we design a 2×2 experimental framework to assess model performance across in-distribution and out-of-distribution image and text inputs. Our results show that appropriate regularization, such as constraining the number of trainable parameters or adopting a low learning rate, effectively prevents forgetting when dealing with out-of-distribution images. However, we uncover a distinct form of forgetting in settings with in-distribution images and out-of-distribution text. We attribute this forgetting as task-specific overfitting and address this issue by introducing a data-hybrid training strategy that combines datasets and tasks. Finally, we demonstrate that this approach naturally extends to continual learning, outperforming existing methods with complex auxiliary mechanisms. In general, our findings challenge the prevailing assumptions by highlighting the inherent robustness of MLLMs and providing practical guidelines for adapting them while preserving their general capabilities.

## 1 INTRODUCTION

The remarkable success of multimodal large language models (MLLMs) in general-purpose visual reasoning (Alayrac et al., 2022; Liu et al., 2023; Achiam et al., 2023) has spurred significant interest in adapting them to specialized downstream applications. Compared to large language models (LLMs), MLLM fine-tuning is not merely beneficial, but often necessary, as visual data presents distinct challenges compared to text. Visual inputs are exceptionally high-dimensional, and many specialized domains are poorly represented in the data used for pre-training. Consequently, out-of-the-box MLLMs can struggle in critical applications, whether it is a robot not able to generalize to unseen rooms (Shi et al., 2025), a web agent misinterpreting novel screenshot layouts (Xie et al., 2024), or a biological application unable to identify specific cell types (Burgess et al., 2025).

However, the prevailing wisdom suggests that fine-tuning MLLMs is risky due to catastrophic forgetting, a phenomenon in which specialization on a new task severely degrades a model's general capabilities (Zhai et al., 2024; Shuttleworth et al., 2024). To address this, previous work has proposed a suite of complex solutions, ranging from sophisticated regularization schemes and parameter isolation techniques to intricate methods (Wang et al., 2023; Shuttleworth et al., 2024; Chen et al., 2023; Li et al., 2025). These approaches often introduce significant architectural or training overhead, reinforcing the notion that preserving general MLLM knowledge is an inherently difficult problem (McCloskey & Cohen, 1989; Andreassen et al., 2021).

Surprisingly, our systematic study reveals that for MLLMs, catastrophic forgetting is largely not a problem. We fine-tune state-of-the-art MLLMs, `Qwen2.5-VL-3B` (Bai et al., 2025), on the ImageNet image classification task and evaluate them on a comprehensive 2x2 matrix, testing performance on both in-distribution (ID) and out-of-distribution (OOD) image and text inputs (§2). Our central finding is that with a simple and proper fine-tuning recipe, such as using a low learning rate or employing parameter-efficient fine-tuning, MLLMs maintain their general-purpose performance, especially when handling OOD visual inputs (§3.1, §3.2). We verify that this conclusion holds across MLLM architectures, including `LLaVA1.5-7B` (Liu et al., 2023) and `Qwen2.5-VL-7B`, as well as in extremely OOD fine-tuning domains, such as surgery and microscopy, challenging the idea that a trade-off between specialization and generalization is inevitable (§3.3).

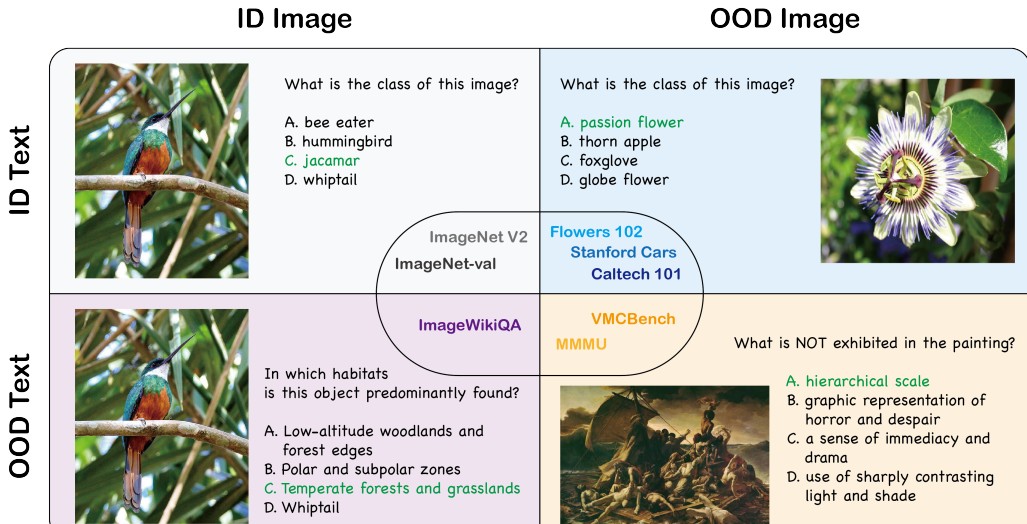

Figure 1: **Evaluation matrix.** A 2×2 design crossing *text* and *image*. In this work, for both text and images, we define in-distribution (ID) data as samples drawn from the same probability distribution as the training set. Conversely, out-of-distribution (OOD) data originates from a distribution not encountered during training; during evaluation, we report average accuracy within each quadrant. This setup allows us to systematically evaluate a comprehensive range of training and evaluation scenarios. Further details on the datasets are provided in Appendix B.1.

However, our investigation revealed one specific and important failure mode (§4.1): forgetting occurs on tasks involving ID images paired with OOD text (e.g., the same ImageNet image but with different questions about the objects than classification). We determine that this scenario reduces the problem to a uni-modal language task; since the images are familiar, the model's behavior is dictated by its language component. Here, the model overfits to the linguistic patterns of the training prompts and fails to follow new instructions at inference time, which we call task-specific overfitting (§4.2). We demonstrate that this issue can be resolved with a simple data-hybrid training strategy, which involves mixing a small amount of general-purpose data with the task-specific fine-tuning dataset to prevent this narrow overfitting (§4.3).

Armed with this complete understanding of MLLM fine-tuning, we extend our findings from single fine-tuning to the challenging continual learning setting (Luo et al., 2025; Chen et al., 2024b). In the newly created continual learning benchmark, which requires the MLLM to learn five challenging remote sensing, medical, autonomous driving, science, and finance knowledge, we show that our straightforward approach allows MLLMs to sequentially learn new tasks while preserving prior knowledge (§5), outperforming all complex methods that rely on mechanisms like data replay buffers (Zhao et al., 2025). This result underscores that the intrinsic capacity of MLLMs for continual learning is much greater than previously understood.

We believe our primary contribution is to reframe the community's understanding of MLLM adaptation. We demonstrate that the perceived threat of catastrophic forgetting has been overstated and that effective, robust fine-tuning can be achieved with a remarkably simple recipe. We hope these findings encourage practitioners to move beyond unnecessarily complex solutions and adopt this parsimonious approach to unlock the full potential of MLLMs in diverse real-world applications.

## 2  MLLM FINE-TUNING: EVALUATION PROTOCOLS AND TRAINING RECIPES

This section specifies *how* we evaluate and *how* we fine-tune multimodal large language models (MLLMs). We first define a controlled protocol built around a 2×2 distribution shift matrix, then describe the models, training setup, and prompting templates used throughout.

### 2.1  EVALUATION PROTOCOLS

**Fine-tuning task and dataset.** We establish a consistent starting point by fine-tuning a multiple-choice visual question answering task constructed from ImageNet, which we call **ImageNet-VQA**.

For each ImageNet image, we pose a single question asking for its class label with four options (A–D): one ground-truth label and three distractors. To increase the difficulty of the fine-tuning task, we employ CLIP (Radford et al., 2021) to select the most challenging distractors, with the methodology detailed in Appendix B.1.

We choose ImageNet because it provides (i) large-scale, diverse, natural images with standardized labels; (ii) a clean mapping to unambiguous multiple-choice questions; and (iii) a familiar in-distribution (ID) reference point for studying shifts in either text or image domains.

**Axes of variation: text and image.** Our evaluation isolates two sources of distribution shift: *Text* (the question form) and *Image* (the visual domain). ID text is the same classification question format used for fine-tuning; OOD text uses question styles that require different reasoning skills or external knowledge. ID images are natural photographs similar to ImageNet; OOD images come from different object sets or visual domains (e.g., flowers or stylized drawings).

**The $2\times2$ evaluation matrix.** Crossing the two axes yields four standardized scenarios (Figure 1):

- **ID Text + ID Image ($\text{ID}^T$–$\text{ID}^I$):** in-distribution questions on in-distribution images. Datasets: ImageNet (Deng et al., 2009) (validation split) and ImageNetV2 (Recht et al., 2019).
- **ID Text + OOD Image ($\text{ID}^T$–$\text{OOD}^I$):** in-distribution questions on out-of-distribution images. Datasets: Flowers102 (Nilsback & Zisserman, 2008), Caltech101 (Fei-Fei et al., 2004), Stanford Cars (Krause et al., 2013).
- **OOD Text + ID Image ($\text{OOD}^T$–$\text{ID}^I$):** novel questions on in-distribution images. Dataset: ImageWikiQA (Zhang et al., 2024).
- **OOD Text + OOD Image ($\text{OOD}^T$–$\text{OOD}^I$):** novel questions on out-of-distribution images. Datasets: MMMU (Yue et al., 2024), VMCBench (Zhang et al., 2025).

Unless otherwise noted, we report the accuracy averaged within each quadrant for clarity.

## 2.2 TRAINING RECIPES

**Base models.** We study two widely used MLLM families, `Qwen2.5-VL` (Bai et al., 2025) and `LLaVA` (Liu et al., 2023). Our main ablations in §3 and §4 use `Qwen2.5-VL-3B`; we additionally validate our findings on `Qwen2.5-VL-7B` and `LLaVA-1.5-7B`. For comparisons on the MLLM-CL benchmark in §5, we adopt `LLaVA-1.5-7B` to align with previous work (Zhao et al., 2025).

**Codebase and hyperparameters.** We train with `LLaMA-Factory` (Zheng et al., 2024). Unless specified, we use a batch size of 16 and ablate the learning rate of $\{1e-5, 1e-6\}$. Training runs for one epoch on ImageNet-VQA (approximately 80,000 steps). We compare different trainable parameters and keep other settings fixed for fair comparison; full configurations are listed in Appendix C. Since **LLM Backbone Fine-tuning** is redundant and not commonly used, we donate the recipe that unfreezing all LLM backbone parameters while freezing all the vision encoder and project parameters as **Full Fine-tuning**.

**Prompts and templates.** We use the system templates provided by `LLaMA-Factory` for `Qwen2.5-VL` and `LLaVA`. All evaluations in §3 and §4 follow the multiple-choice format. To avoid formatting confounding, the prompts explicitly instruct the model to output a single option letter (A–D). Illustrative prompt templates are included in Appendix D.1.

# 3 FINE-TUNING WITHOUT FORGETTING: A SIMPLE RECIPE WITHOUT PERFORMANCE TRADE-OFF

Can a multimodal large language model (MLLM) be specialized to a new task *without* erasing its general capabilities? Using the $2\times2$ evaluation matrix (§2), we vary the trainable components (LLM backbone, vision encoder, projector), optimization method (full fine-tuning vs. LoRA), and learning rate.

Three consistent findings emerge: (I) with simple regularization (small learning rate or LoRA), forgetting on OOD images is *nearly absent* as ID accuracy increases; (II) avoiding forgetting does *not* reduce target-task accuracy; and (III) these patterns hold across model sizes/families, rare visual domains, and low-data regimes.

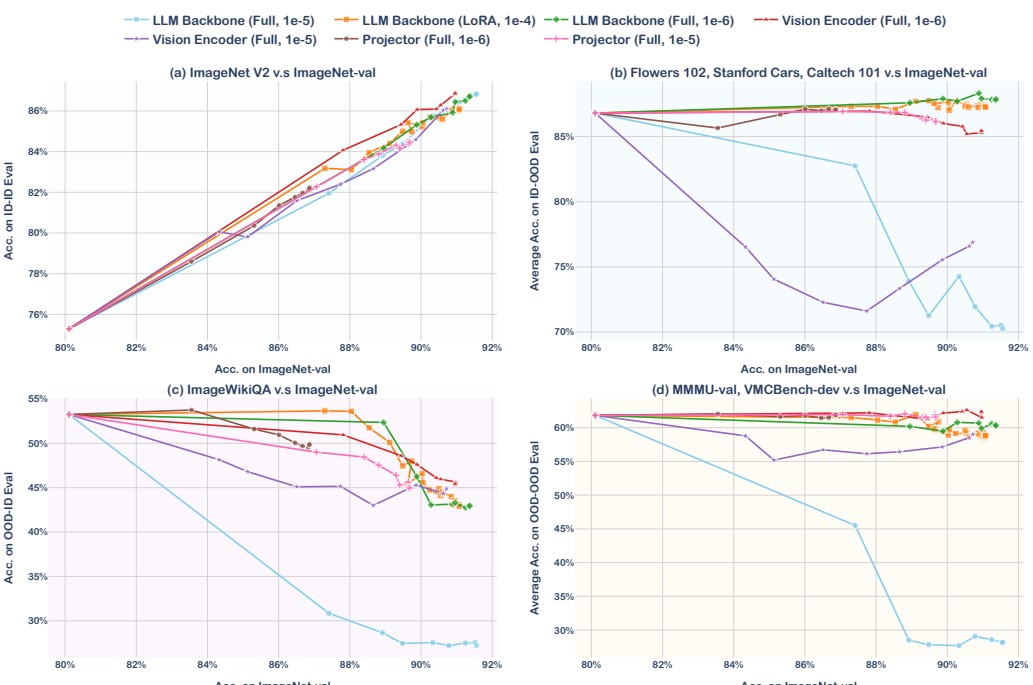

Figure 2: **Single-task fine-tuning across the evaluation matrix.** Each curve traces checkpoints during fine-tuning: x-axis = ID accuracy on ImageNet validation (the fine-tuned task), y-axis = accuracy on an ID/OOD evaluation. Layout and colors follow Figure 1. Legends show **trainable part (method, learning rate)**. Performance is largely maintained in $\mathbf{ID}^T$–$\mathbf{OOD}^I$ and $\mathbf{OOD}^T$–$\mathbf{OOD}^I$ with simplest regularization on parameter updata, with a notable drop only in $\mathbf{OOD}^T$–$\mathbf{ID}^I$. Full hyperparameters are in Appendix C.1.

## 3.1 FINDING I: SIMPLE REGULARIZATION PREVENTS (NEARLY ALL) FORGETTING

**Research question.** Catastrophic forgetting is often attributed to architectural limits: specializing on a new task is thought to overwrite broad, pre-trained knowledge. If that were the case, the gains on the ID data should come with the losses on the OOD data.

**Results.** In Figure 2, high-learning-rate full fine-tuning (1e-5) increases ID accuracy but substantially degrades OOD performance, consistent with catastrophic forgetting: relative to zero-shot, **LLM Backbone, Full, 1e-5** yields $-16.56$ pp on $\mathbf{OOD}^T$–$\mathbf{ID}^I$ and $-33.64$ pp on $\mathbf{OOD}^T$–$\mathbf{OOD}^I$ (Table 1). In contrast, conservative settings (small learning rate or LoRA) keep the OOD accuracy essentially stable as the ID accuracy increases. Restricting the magnitude and scope of parameter updates eliminates these drops: **LLM Backbone, Full, 1e-6** changes are $+1.06$ pp ($\mathbf{OOD}^T$–$\mathbf{ID}^I$) and $-1.51$ pp ($\mathbf{OOD}^T$–$\mathbf{OOD}^I$); **LLM Backbone, LoRA, 1e-4** changes are $+0.46$ pp and $-2.97$ pp, respectively.

> *Takeaway 1:* Forgetting is *not* inevitable; it arises from over-optimization. Simple regularization (small learning rate or parameter-efficient training) preserves capabilities.

## 3.2 FINDING II: NO TRADE-OFF BETWEEN SPECIALIZATION AND PRESERVATION

**Research question.** Prior reports suggest a performance gap between full fine-tuning and LoRA on the target task. If regularization preserves OOD performance, does it *cost* ID accuracy?

**Results.** Table 1 shows that the regularized settings match the aggressive baseline on the ID task while avoiding OOD forgetting. Validation accuracy differences relative to **LLM Backbone, Full, 1e-5** are $\leq 0.6$pp for **LLM Backbone, Full, 1e-6** ($-0.19$pp), **LLM Backbone, LoRA, 1e-4** ($-0.48$pp), and **Vision Encoder, Full, 1e-6** ($-0.60$pp). Projector-only fine-tuning is the sole exception (e.g., $-4.70$pp at 1e-6) and is therefore not recommended when target-task accuracy is critical.

| Trainable Part | Settings | Final Acc. | $\Delta$ vs. zero-shot | |
| | | Validation (%) | $\text{OOD}^T$–$\text{ID}^I$ (pp) | $\text{OOD}^T$–$\text{OOD}^I$ (pp) |
|---|---|---|---|---|
| LLM Backbone | Full, 1e-5 | 91.56 | -16.56 | -33.64 |
| LLM Backbone | LoRA, 1e-4 | 91.08 (-0.48) | 0.46 | -2.97 |
| LLM Backbone | Full, 1e-6 | 91.37 (-0.19) | 1.06 | -1.51 |
| Vision Encoder | Full, 1e-6 | 90.96 (-0.60) | -1.36 | 0.49 |
| Vision Encoder | Full, 1e-5 | 91.08 (-0.48) | -9.90 | 2.76 |
| Projector | Full, 1e-6 | 86.86 (-4.70) | 0.26 | 0.05 |
| Projector | Full, 1e-5 | 89.68 (-1.88) | -0.64 | -0.26 |

Table 1: **ID accuracy and robustness deltas across recipes.** "*Final Acc*" is ImageNet-VQA validation accuracy; in parentheses we show the difference to **LLM Backbone, Full, 1e-5**. "$\Delta$ *vs. zero-shot*" reports percentage-point change relative to the pre-trained model on $\text{OOD}^T$–$\text{ID}^I$ and $\text{OOD}^T$–$\text{OOD}^I$. To enhance visual clarity, we use red to highlight performance degradations $>$3pp and blue for changes within a $\pm$3pp margin. Rows corresponding to settings where all results fall within this margin are shaded gray . This suggests that most of regularization strategies mitigate catastrophic forgetting without compromising the model's learning capacity.

(a) Model size and family.

| Model Version | Validation (%) | ImageNetV2 (%) | ID–OOD (%) | OOD–OOD (%) |
|---|---|---|---|---|
| Qwen2.5-VL-3B | 80.11→91.37 | 75.29→86.72 | 86.80→87.87 | 61.82→60.31 |
| Qwen2.5-VL-7B | 83.20→92.66 | 78.61→88.05 | 90.35→91.24 | 62.57→62.62 |
| LLaVA-7B | 65.53→91.43 | 61.55→86.76 | 66.44→70.05 | 41.45→37.73 |

(b) Rare domains.

| Dataset | Validation (%) | OOD–OOD (%) |
|---|---|---|
| ImageNet | 80.11→89.88 | 61.82→59.48 |
| BSCCM | 18.15→84.34 | 61.82→61.19 |
| PitVis | 25.61→51.33 | 61.82→61.56 |

(c) Dataset size.

| Dataset fraction | Validation (%) | |
|---|---|---|
| | lr=1e-6 | lr=1e-5 |
| 100% | 91.42 | 91.60 |
| 25% | 90.18 | 89.08 |
| 2.5% | 86.99 | 87.46 |
| 0.25% | 82.03 | 81.82 |

Table 2: **Generalization of the recipe.** The default setting referenced in §3.2 is shaded in gray . The results show that all findings in §3.1 are consistent across: (a) different model sizes and families; (b) rare domains including surgery and microscopy; (c) different fine-tuning datasets size; Full training details are in Appendix C.2.

> **Takeaway 2:** Specialization and preservation are *not* at odds: Under regularized fine-tuning, ID and OOD performance do not trade off.

## 3.3 FINDING III: CONSISTENCY ACROSS MODELS, DOMAINS, AND DATA REGIMES

**Research question.** If the recipe is principled, it should transfer across architectures, uncommon visual domains, and data-scarce settings.

**Results.** *Models.* The trends persist across sizes and families (Table 2a): Qwen2.5-VL-3B improves ImageNet validation $80.11 \to 91.37$ with $\text{OOD}^T$–$\text{OOD}^I$ $61.82 \to 60.31$ $(-1.51\text{pp})$; Qwen2.5-VL-7B improves $83.20 \to 92.66$ with $\text{OOD}^T$–$\text{OOD}^I$ $+0.05$pp; LLaVA-1.5-7B improves $65.53 \to 91.43$ with a modest $\text{OOD}^T$–$\text{OOD}^I$ drop $(-3.72\text{pp})$.

*Rare domains.* The same recipe holds for microscopy (BSCCM (Pinkard et al., 2024)) and surgical (PitVis (Das et al., 2025)) data (Table 2b), keeping $\text{OOD}^T$–$\text{OOD}^I$ within $\leq 2.5$pp while yielding large ID gains (+66pp on BSCCM, +26pp on PitVis).

*Data size.* Even at $0.25\%$ of the data, a small learning rate (1e-6) remains competitive in the ID task (82.03 vs. 81.82 at 1e-5; Table 2c).

> **Takeaway 3:** These findings generalize across architectures, domains, and data regimes, implying that forgetting in MLLM fine-tuning is generally not a concern.

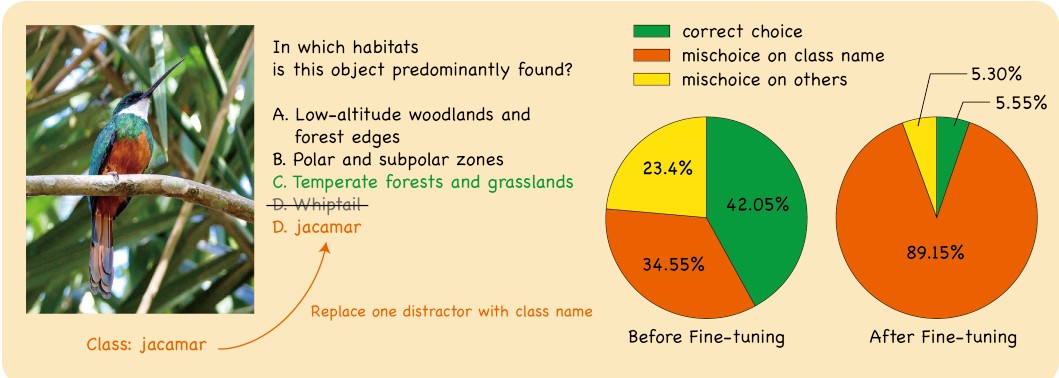

Figure 3: **ImageWikiQA with class-label distractors.** *Left:* an example transformation where one distractor is replaced by the correct class name. *Right:* accuracy with/without a class-name distractor, before fine-tuning and after fine-tuning, using **LLM Backbone, Full, 1e-6**. The substantial decrease in accuracy and the concurrent increase in "mischoice on class name" after fine-tuning indicate that the model ceases to follow prompt instructions, instead defaulting to outputting the choice with class label directly. Therefore, the primary issue is *task-specific overfitting* rather than catastrophic forgetting.

## 4 OOD TEXT MEETS ID IMAGES: DIAGNOSIS AND SIMPLE REMEDY

Our $2\times2$ evaluation reveals a single weak spot: $\text{OOD}^T$–$\text{ID}^I$ (novel text over familiar images), exemplified by ImageWikiQA. In contrast to $\text{ID}^T$–$\text{OOD}^I$ and $\text{OOD}^T$–$\text{OOD}^I$, where regularization preserves performance, Figure 2c shows a clear drop on OOD text with ID images. We (i) diagnose this failure as *task-specific overfitting* in the ID image distribution and (ii) demonstrate that a simple *data-hybrid* recipe prevents it with minimal impact on the target task.

### 4.1 FINDING IV: FORGETTING APPEARS ONLY WITH OOD TEXT OVER ID IMAGES

**Research question.** In $\text{OOD}^T$–$\text{ID}^I$, the image distribution matches fine-tuning (ID), but the text distribution shifts. The test set, ImageWikiQA (Zhang et al., 2024), asks the model to link an ImageNet image to external knowledge (e.g., the habitat of a species or the use of an artifact) rather than to perform the ImageNet classification task. This setup closely parallels standard LLM fine-tuning, where inputs remain in-domain while the instruction distribution changes. Prior work on LLMs has shown that single-task fine-tuning can impair other capabilities and encourage instruction-ignoring (Luo et al., 2025; Ung et al., 2024; Lyu et al., 2024).

**Results.** Even with regularized fine-tuning (e.g., small learning rates or LoRA), ImageWikiQA performance drops relative to zero-shot (Figure 2c). For example, the **LLM Backbone, Full, 1e-6** configuration falls from 53.35% to 42.95% ($-10.40$pp) after fine-tuning on ImageNet. This contrasts sharply with $\text{ID}^T$–$\text{OOD}^I$ and $\text{OOD}^T$–$\text{OOD}^I$, where performance remains stable under the same settings.

> *Takeaway 4:* The sole exception in our study is the ID-image/OOD-text setting, where forgetting persists and is not remedied by standard regularized fine-tuning, mirroring findings from LLM fine-tuning.

### 4.2 FINDING V: $\text{OOD}^T$–$\text{ID}^I$ FORGETTING ARISES FROM TASK-SPECIFIC OVERFITTING

**Research question.** We hypothesize that the model becomes over-attuned to the "classify-this-image" template when trained on ID images. To test this, we construct **ImageWikiQA with class-label distractors** by replacing one standard distractor with the correct class label (Figure 3, left). If the model has memorized the task, it should over-select the class label rather than the correct answer.

**Results.** Using the **LLM Backbone, Full, 1e-6** model, we observe severe *task-specific overfitting*: *before* fine-tuning, accuracy drops moderately when the class-name distractor is present (53.25% $\rightarrow$

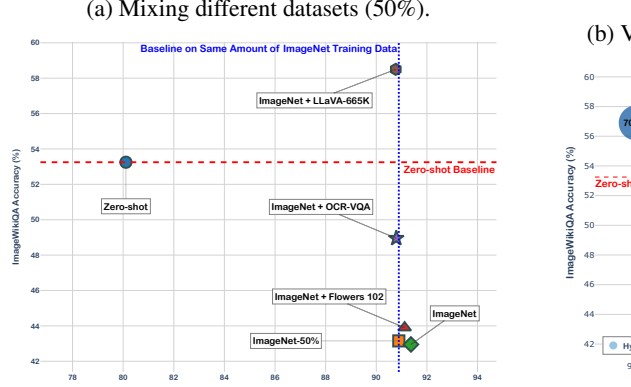

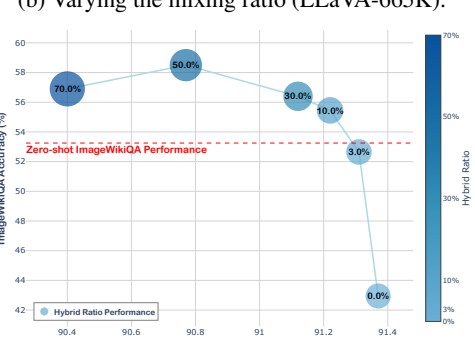

Figure 4: **Ablations for data-hybrid training.** (a) Mixing ImageNet-VQA with Flowers102, OCR-VQA, or LLaVA-665K (each at 50% of training instances). (b) Varying the LLaVA-665K mix from 0% to 70%; larger, darker markers denote higher ratios. Augmenting the training data with diverse textual inputs helps to alleviate *task-specific overfitting*. Consequently, this data-hybrid method improves model robustness in the $\mathbf{OOD}^T$–$\mathbf{ID}^I$ setting with minimal trade-offs for ID performance. Training details are in Appendix C.3.

42.05%, $-11.2$ pp); *after* fine-tuning, the drop is drastic ($42.95\% \rightarrow 5.55\%$, $-37.4$ pp) (Figure 3, right). The much larger change after fine-tuning indicates a learned bias to "pick the class label,", that is, prompt-ignoring rather than knowledge deletion.

---

> **Takeaway 5:** Forgetting in the ID-image/OOD-text case stems from *task-specific overfitting*: the model memorizes the image-specific classification template during fine-tuning and ignores the prompt.

---

## 4.3 FINDING VI: DATA-HYBRID TRAINING PREVENTS TASK OVERFITTING

**Research question.** If overfitting arises from repeatedly pairing ID images with a single classification template, mixing in *diverse* tasks should force the model to attend to the prompt and avoid the shortcut. We therefore ablate both **dataset type** and **mixing ratio**.

**Results.** *Dataset type (50% mix).* Figure 4a compares mixing ImageNet-VQA with: (i) Flowers102 (ID-style text on OOD images), (ii) OCR-VQA (OOD text), and (iii) LLaVA-665K (broad OOD instructions). Hybrid training consistently improves $\mathbf{OOD}^T$–$\mathbf{ID}^I$ while keeping ImageNet-VQA strong. Flowers102 yields only marginal gains on ImageWikiQA (another classification-style dataset, hence weak against task overfitting). OCR-VQA helps more by requiring text-based reasoning. LLaVA-665K performs best, likely due to its breadth of instructions and reasoning styles.

*Mixing ratio (with LLaVA-665K).* Figure 4b shows that increasing the proportion of LLaVA-665K to 50% keeps ImageNet-VQA within $\sim$1 pp of the pure-ImageNet condition while markedly improving ImageWikiQA; at 70%, we see no further $\mathbf{OOD}^T$–$\mathbf{ID}^I$ gains. This suggests that the effect is not just "more data," but specifically *task diversity* mitigating overfitting.

Finally, the effectiveness of co-training with OCR-VQA and LLaVA-665K indicates that, although overfitting manifests on ID images, the remedy does not require additional ID images. Greater task diversity alone is sufficient to counteract the bias, regardless of the image distribution. In addition, we have also shown that the synthetic data (LLaVA-665K) is effective, which furthermore provides a positive result on the robustness of hybrid training.

---

> **Takeaway 6:** Data-hybrid fine-tuning—mixing diverse instruction data (without requiring ID images)—preserves ID-task accuracy while overcoming ID-image/OOD-text forgetting.

---

| Method | RS (%) | | Med (%) | | AD (%) | | Sci (%) | | Fin (%) | |
|---|---|---|---|---|---|---|---|---|---|---|
| | *Last* | *Average* | *Last* | *Average* | *Last* | *Average* | *Last* | *Average* | *Last* | *Average* |
| Zero-shot | 32.29 | - | 28.28 | - | 15.59 | - | 35.55 | - | 62.56 | - |
| *w/ replay buffer* | | | | | | | | | | |
| LoRA | 29.57 | 80.87 | 29.19 | 58.60 | 7.09 | 38.95 | 19.55 | 36.41 | 63.60 | 36.78 |
| MoELoRA | 40.23 | 80.00 | 23.58 | 56.91 | 5.19 | 34.69 | 18.35 | 31.70 | 74.89 | 31.36 |
| O-LoRA | 76.21 | 80.13 | 51.34 | 70.23 | 36.50 | 61.35 | 42.64 | 53.34 | 90.20 | 59.38 |
| L2P | 75.21 | 80.09 | 38.50 | 68.64 | 32.31 | 54.79 | 41.05 | 48.68 | 88.05 | 55.02 |
| ModalPrompt | 64.77 | 80.11 | 38.60 | 60.99 | 20.61 | 50.67 | 29.98 | 41.97 | 88.22 | 48.44 |
| HiDe-LLaVA | 75.36 | **81.51** | 39.23 | 62.37 | 37.17 | 49.37 | 45.02 | 50.61 | 81.89 | 55.73 |
| MR-LoRA | **79.87** | 80.82 | **62.71** | **72.19** | 51.89 | 65.41 | 52.48 | 62.52 | 89.69 | **67.31** |
| **IncLoRA (Ours)** | 77.43 | 78.30 | 62.57 | 71.93 | **52.00** | 65.38 | 52.48 | 62.12 | 90.41 | 66.98 |
| **SeqFull (Ours)** | 78.94 | 75.62 | 62.45 | 72.16 | 51.50 | **65.77** | 52.08 | 62.32 | **91.21** | 67.24 |
| *w/o replay buffer* | | | | | | | | | | |
| LoRA | 26.75 | 80.72 | 25.76 | 59.68 | 0.79 | 40.51 | 18.69 | 18.64 | 70.44 | 28.49 |
| MoELoRA | 21.42 | 80.05 | 25.29 | 57.26 | 0.79 | 37.03 | 17.01 | 19.65 | 60.34 | 24.97 |
| O-LoRA | 62.68 | 80.22 | 35.17 | 67.56 | 16.93 | 51.51 | 34.44 | 44.28 | 92.16 | 48.28 |
| L2P | 63.82 | 80.02 | 34.63 | 68.86 | 22.96 | 51.57 | 38.58 | 45.12 | **92.98** | 50.59 |
| ModalPrompt | 65.99 | 80.11 | 37.35 | 59.66 | 23.27 | 46.86 | 37.61 | 42.97 | 87.60 | 50.36 |
| HiDe-LLaVA | 41.17 | **80.91** | 30.33 | 65.47 | 18.73 | 39.78 | 37.08 | 32.92 | 92.21 | 43.90 |
| **IncLoRA (Ours)** | 77.20 | 77.59 | 58.97 | 71.59 | 51.43 | 64.40 | 47.44 | 60.22 | 90.24 | 65.06 |
| **SeqFull (Ours)** | **79.10** | 77.06 | **61.22** | **72.75** | **52.36** | **66.09** | 50.52 | 62.49 | 91.29 | **67.44** |

Table 3: **Continual learning on the MLLM-CL benchmark.** We highlight **best** and second best separately for *with* and *without* replay. Our simple methods (IncLoRA, SeqFull) are competitive with specialized approaches under replay, and dominate most columns without replay.

## 5 FROM SINGLE TO MULTIPLE: SIMPLE STRATEGIES RIVAL SOTA

Our single-task study shows that catastrophic forgetting can be substantially reduced with regularization (§3) and data hybrid training (§4). The natural question is whether these observations carry over from one task to a sequence of tasks. We therefore turn to *continual learning*, where a model learns tasks one after another while preserving performance on earlier tasks. Perhaps unexpectedly, we find that very simple updates, either LoRA or a small learning rate, match or outperform prior methods purpose-built for continual learning, both *with* and *without* a replay buffer.

### 5.1 BENCHMARK AND EVALUATION

**Benchmark.** We use the MLLM continual learning benchmark introduced by MLLM-CL (Zhao et al., 2025), spanning five domains in a fixed order: **R**emote **S**ensing → **Med**icine → **A**utonomous **D**riving → **Sci**ence → **Fin**ance. See §B.3 for more details.

**Evaluation.** Continual learning reframes forgetting from "does fine-tuning erase zero-shot skills?" to "does learning the next task erase the previous one?". We therefore report two standard metrics: *Last* (performance on each task after training on the full sequence) and *Average* (mean performance across tasks at the time each task is learned). Details appear in §D.4.

**Experimental setup.** For comparability, we follow the MLLM-CL recipe exactly (optimizer, prompts, and models), adopt their evaluation protocol, and use the same dataset splits. The zero-shot row in Table 3 provides the pre-training baseline before any fine-tuning.

### 5.2 FINDING VII: SIMPLE STRATEGIES COMPETE WITH SOTA IN CONTINUAL LEARNING

**Method.** We evaluate two simple continual learning strategies: incremental LoRA (**IncLoRA**) and sequential full fine-tuning (**SeqFull**). For **IncLoRA**, we train a new LoRA *adapter* for each task and, after training, merge the adapter weights into the base model, which then initializes the next task. **SeqFull** simply fine-tunes all model parameters for each task in sequence, without additional mechanisms.

We refer our **IncLoRA** and **SeqFull** as *simple* because all the other method shown in Table 3 either use a router to select an appropriate LoRA instead of merging, or add additional regularization during fine-tuning the new LoRA. Our framework is essentially a simplified version of them.

**Results.** With a replay buffer (a bounded memory that retains a small sample of past tasks' examples and replays them alongside the current task's data to reduce catastrophic forgetting), many prior methods introduce sophisticated components to control forgetting, yet our simple approaches achieve performance comparable to state-of-the-art techniques. For example, **SeqFull** attains 78.94% on **RS** under the *Last* metric, closely matching **MR-LoRA** (79.87%) while outperforming it in **Fin**.

The gap widens in the more restrictive no-replay setting, which is important for privacy-sensitive applications (e.g., medicine) where replay is infeasible. Except for the *Average* metric in the first task (**RS**) and the *Last* metric on the final task (**Fin**)—both of which do not reflect forgetting—**IncLoRA** and **SeqFull** outperform all competing methods in the remaining eight comparisons, establishing new state-of-the-art results in most domains.

> *Takeaway 7:* Simple update policies rival or exceed specialized continual-learning methods, work in privacy-sensitive no-replay settings, and avoid additional complexity.

## 6 RELATED WORK

**Vision language models.** Multimodal large language models (MLLMs) such as Flamingo (Alayrac et al., 2022), LLaVA (Liu et al., 2023), and GPT-4V (Achiam et al., 2023) demonstrate strong visual–linguistic understanding and reasoning (Xu et al., 2024). A typical MLLM couples a vision encoder with a language backbone—through a projector or cross-attention module—is trained in large image-text corpora and is subsequently adjusted to instruction (Liu et al., 2023). Recent work has emphasized scaling, architectural refinements, and training strategies to improve zero-/few-shot generalization Tong et al. (2024); Chen et al. (2024c); Bai et al. (2025). In this work, we study how to adapt strong base MLLMs to diverse downstream tasks while preserving zero-shot performance, a problem that is arguably more acute for MLLMs than for LLMs, yet comparatively underexplored.

**Catastrophic forgetting.** Catastrophic forgetting is the loss of previously acquired knowledge when a model is trained on new tasks (Kemker et al., 2018; Chen & Liu, 2022; Goodfellow et al., 2013). In LLMs, catastrophic forgetting has been extensively studied—empirically (Kalajdzievski, 2024; Scialom et al., 2022), theoretically (Shuttleworth et al., 2024), methodologically (Chen et al., 2023; Li et al., 2025), and from an evaluation point of view (Ung et al., 2024). In contrast, catastrophic forgetting in MLLMs has received less attention (Zhai et al., 2024). Previous work always shows a result of learning less and forgetting less, while we are presenting the phenomenon of learning the same amount without forgetting.

**Continual learning.** Continual learning aims to acquire new capabilities without erasing prior knowledge (Wang et al., 2024; Chen & Liu, 2022; Hadsell et al., 2020). It is critical in real-world settings where data distributions and taxonomies evolve, centralized retraining may be impractical due to cost or privacy, and preserving generalist abilities (e.g., zero-shot performance) is important for safety and robustness. To mitigate forgetting, previous work explores replay, regularization, and parameter isolation approaches, but these often add considerable compute, memory, and engineering complexity (Zhao et al., 2025; Van de Ven & Tolias, 2019). Although continual learning for MLLMs has begun to be explored (Chen et al., 2024a; Huang et al., 2024), we show that—with appropriate training recipes—forgetting can be largely mitigated, yielding state-of-the-art results with simple and compute-efficient methods.

## 7 CONCLUSION

By rethinking and re-evaluating the design space of multimodal adaptation, this paper reframes how to fine-tune multimodal large language models. We find that concerns about catastrophic forgetting are often overstated. In practice, a simple recipe—using small learning rates or parameter-efficient updates—yields specialized models that remain strong generalists. Our analysis isolates a single failure mode: overfitting to linguistic patterns rather than visual content. We address this with a straightforward hybrid-data mix. On a challenging continual learning benchmark, this recipe performs on par with or better than more complex alternatives, suggesting that vision language models are more intrinsically robust than commonly assumed. We hope these results encourage simpler, more transparent adaptation methods and provide a stable foundation for future work.

## ETHICS STATEMENT

We, the authors of this work, confirm our adherence to the ICLR Code of Ethics. Our research is primarily methodological in nature and does not raise significant ethical concerns regarding data privacy, fairness, or potential misuse, as it does not involve sensitive datasets or direct real-world applications involving individuals.

## REPRODUCIBILITY STATEMENT

To ensure the full reproducibility of our work, we provide our code, adapted datasets, and detailed hyperparameter specifications, which include all scripts to generate the necessary configuration files and perform the training and evaluations presented in this paper.

**Code**: `https://anonymous.4open.science/r/VLM-Forgetting-C1CE/`.

**Datasets**: `https://huggingface.co/datasets/VLM-Forgetting/vlm-forgetting-datasets`.

**Hyperparameters**: Appendix C.

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

## A  Statement on the Use of Large Language Models (LLMs)

In the preparation of this manuscript, the authors utilized a Large Language Model (LLM) as a general-purpose writing assistance tool. The LLM's role was strictly limited to improving the clarity, grammar, and readability of the text. Specific tasks included rephrasing sentences for better flow, correcting grammatical errors, and ensuring consistent terminology.

## B  Datasets Details

### B.1  2x2 Evaluation Matrix Details

**Classification Datasets (Deng et al., 2009; Nilsback & Zisserman, 2008; Fei-Fei et al., 2004; Krause et al., 2013).**  For all classification datasets in the evaluation matrix, we follow the same protocol to turn them into multiple-choice questions. The question text are fixed to *What is the class of this image? Please answer with a single letter (A, B, C, or D).*, where the formatting instructions are concatenated to ensure the evaluation result will not be greatly influenced by output format of the model.

To increase the difficulty of the task and test the model's fine-grained discrimination ability, distractors are strategically selected. We use CLIP (Radford et al., 2021) to identify the five incorrect class labels with the highest semantic similarity scores to the image. From this pool of five candidates, we randomly sample three to serve as distractors. This methodology ensures that incorrect options are semantically plausible, requiring the model to perform a more precise identification. By fine-tuning on ImageNet-VQA, the model is trained to perform a standard, in-distribution (ID) image classification task.

**ImageWikiQA (Zhang et al., 2024).**  Since the ImageWikiQA dataset is already in a format of multple-choice question, we directly use adapt it.

**MMMU and VMCBench (Yue et al., 2024; Zhang et al., 2025).**  Since the MMMU and VMCBench datasets are already in a format of multple-choice question, we directly use adapt them. For all the numbers reported in this paper, we use the MMMU-val split for the evaluation.

### B.2  Rare Datasets Details

**BSCCM.**  We use the original BSCCM (Pinkard et al., 2024) dataset and follow the official guide at `https://github.com/Waller-Lab/BSCCM/blob/main/Getting_started.ipynb` to create a classification question-answering dataset. We collect images from all 23 available channels, including:

- Brightfield
- DF_50, DF_50_Bottom, DF_50_Right,
- DF_55,
- DF_60, DF_60_Bottom, DF_60_Right,
- DF_65,
- DF_70, DF_70_Bottom, DF_70_Right,
- DF_75,
- DF_80, DF_80_Bottom, DF_80_Right,
- DF_85,
- DF_90,
- DPC_Bottom, DPC_Left, DPC_Right, DPC_Top,
- LED119

There are 10 classes in total, and for each question we ask the model to choose from 6 possible choices. The 5 distractors are randomly sampled from all possible choices and we provide the list of classes as follows:

| | | |
|---|---|---|
| 1. neutrophil | 5. basophil | 9. b_lymphocyte |
| 2. nk_lymphocyte | 6. monocyte | 10. t_lymphocyte |
| 3. eosinophil | 7. plasma_cell | |
| 4. lymphocyte | 8. blast_cell | |

To increase the type of questions, we provide multiple choices of prompt while all of them are sharing the same semantic meaning.

- What type of white blood cell is shown in this {*channel_type*} microscopy image?
- Based on the morphological features visible in this {*channel_type*} image, what is the cell type?
- What is the most likely classification of this blood cell captured with {*channel_type*} illumination?
- Which white blood cell type does this {*channel_type*} image represent?
- What type of immune cell is depicted in this {*channel_type*} microscopy image?
- Looking at the cell morphology in this {*channel_type*} image, which cell type is this?
- What is the identity of this cell captured using {*channel_type*} in LED array microscopy?

We provide the following samples in Table 4 from curated dataset. During training and inference, a prompt of "Please answer with a single letter (A, B, C, D, E or F)" is appended at the end to avoid the influence from model response formatting.

Table 4: VQA Dataset Curated from BSCCM

| Image | Question | Choices |
|---|---|---|
|  | What is the identity of this cell captured using brightfield in LED array microscopy? | A. eosinophil
B. neutrophil
C. t_lymphocyte
D. plasma_cell
E. debris_or_artifact
F. unclassified_cell |
|  | What type of white blood cell is shown in this dark field (50 illumination) microscopy image? | A. plasma_cell
B. nk_lymphocyte
C. b_lymphocyte
D. neutrophil
E. unclassified_cell
F. debris_or_artifact |
|  | Based on the morphological features visible in this differential phase contrast (left illumination) image, what is the cell type? | A. basophil
B. unclassified_cell
C. debris_or_artifact
D. t_lymphocyte
E. blast_cell
F. lymphocyte |

**PitVis.** We use the PitVis Challenge (Das et al., 2025) to create a classification dataset aiming to categorize the frame sampled from video according to the surgical instrument appeared. We fix the sample rate to be 1 out of every 6 frames. The total 21 instrument classes are as follows.

Fixed choices:

1. no_secondary_instrument
2. out_of_patient
3. no_visible_instrument/occluded_image_inside_patient

Other choices:

| | | |
|---|---|---|
| 1. bipolar_forceps | 7. irrigation_syringe | 13. ring_curette |
| 2. cottle | 8. kerrisons | 14. spatula_dissector |
| 3. cup_forceps | 9. micro_doppler | 15. stealth_pointer |
| 4. dural_scissors | 10. nasal_cutting_forceps | 16. suction |
| 5. freer_elevator | 11. pituitary_rongeurs | 17. surgical_drill |
| 6. haemostatic_foam | 12. retractable_knife | 18. tissue_glue |

We still ask the model to choose from 6 possible choices. For every question, there will be 3 fixed choices to be *no_secondary_instrument, out_of_patient, no_visible_instrument* and we will randomly sample 2 or 3 distractors from all other classes (2 if the ground truth is not one of the 3 fixed classes).

We provide the following samples in Table 5 from curated dataset. During training and inference, a prompt of "Please answer with a single letter (A, B, C, D, E or F)" is appended at the end to avoid the influence from model response formatting.

Table 5: VQA Dataset Curated from BSCCM

| Image | Question | Choices |
|---|---|---|
|  | What is the major surgical instrument being used in this frame? | A. tissue_glue
B. retractable_knife
C. haemostatic_foam
D. no_secondary_instrument
E. out_of_patient
F. no_visible_instrument/occluded_image_inside_patient |
|  | What is the major surgical instrument being used in this frame? | A. plasma_cell
B. nk_lymphocyte
C. b_lymphocyte
D. neutrophil
E. unclassified_cell
F. debris_or_artifact |
|  | What is the major surgical instrument being used in this frame? | A. out_of_patient
B. ring_curette
C. no_visible_instrument/occluded_image_inside_patient
D. freer_elevator
E. micro_doppler
F. no_secondary_instrument |

## B.3 MLLM-CL DETAILS

This sequential learning benchmark MLLM-CL contains:

- **RS**: Remote Sensing Data **RSVQA** (60k Training Data)
- **Med**:Medical Data **PathVQA** (23k Training Data)
- **AD**:Auto-Driving Data **DriveLM** (60k Training Data)
- **Sci**:Science Data **AI2D, SciVerse, MapQA, TQA** (33k Training Data)
- **Fin**:Financial Data **StockQA** (60k Training Data).

More details about the dataset can be found in MLLM-CL paper (Zhao et al., 2025). We adapt the number reported in original MLLM-CL paper, including LoRA Hu et al. (2022), MoELoRA Chen et al. (2024a), O-LoRA Wang et al. (2023), L2P Wang et al. (2022), ModalPrompt Zeng et al. (2024), HiDe-LLaVA* Guo et al. (2025), MR-LoRA Zhao et al. (2025)

# C  TRAINING HYPER-PARAMETERS AND DETAILS

## C.1  TRAINING HYPER-PARAMETERS FOR FIGURE 2

In this section, we align the table caption with Figure 2.

| Config | Value |
|---|---|
| Optimizer | AdamW |
| Batch Size | 32 |
| Learning Rate Schedule | cosine decay |
| Warmup Ratio | 0.1 |
| Learning Rate | $1 \times 10^{-4}$ |
| Training Steps | 40000 |
| LoRA Rank | 8 |
| Freeze Vision Tower | True |
| Freeze Multi Modal Projector | True |
| Freeze Language Model | False |

(a) LLM Backbone (LoRA, 1e-4)

| Config | Value |
|---|---|
| Optimizer | AdamW |
| Batch Size | 16 |
| Learning Rate Schedule | cosine decay |
| Warmup Ratio | 0.1 |
| Learning Rate | $1 \times 10^{-5}$ |
| Training Steps | 80000 |
| Freeze Vision Tower | True |
| Freeze Multi Modal Projector | True |
| Freeze Language Model | False |

(b) LLM Backbone (Full, 1e-5)

| Config | Value |
|---|---|
| Optimizer | AdamW |
| Batch Size | 16 |
| Learning Rate Schedule | cosine decay |
| Warmup Ratio | 0.1 |
| Learning Rate | $1 \times 10^{-6}$ |
| Training Steps | 80000 |
| Freeze Vision Tower | True |
| Freeze Multi Modal Projector | True |
| Freeze Language Model | False |

(c) LLM Backbone (Full, 1e-6)

| Config | Value |
|---|---|
| Optimizer | AdamW |
| Batch Size | 16 |
| Learning Rate Schedule | cosine decay |
| Warmup Ratio | 0.1 |
| Learning Rate | $1 \times 10^{-5}$ |
| Training Steps | 80000 |
| Freeze Vision Tower | False |
| Freeze Multi Modal Projector | True |
| Freeze Language Model | True |

(d) Vision Encoder (Full, 1e-5)

| Config | Value |
|---|---|
| Optimizer | AdamW |
| Batch Size | 16 |
| Learning Rate Schedule | cosine decay |
| Warmup Ratio | 0.1 |
| Learning Rate | $1 \times 10^{-6}$ |
| Training Steps | 80000 |
| Freeze Vision Tower | False |
| Freeze Multi Modal Projector | True |
| Freeze Language Model | True |

(e) Vision Encoder (Full, 1e-6)

| Config | Value |
|---|---|
| Optimizer | AdamW |
| Batch Size | 16 |
| Learning Rate Schedule | cosine decay |
| Warmup Ratio | 0.1 |
| Learning Rate | $1 \times 10^{-5}$ |
| Training Steps | 80000 |
| Freeze Vision Tower | True |
| Freeze Multi Modal Projector | False |
| Freeze Language Model | True |

(f) Projector (Full, 1e-5)

| Config | Value |
|---|---|
| Optimizer | AdamW |
| Batch Size | 16 |
| Learning Rate Schedule | cosine decay |
| Warmup Ratio | 0.1 |
| Learning Rate | $1 \times 10^{-6}$ |
| Training Steps | 80000 |
| Freeze Vision Tower | True |
| Freeze Multi Modal Projector | False |
| Freeze Language Model | True |

(g) Projector (Full, 1e-6)

## C.2 Training Hyper-parameters for Table 2

In the ablation across different setting, we study the fine-tuning recipt of full fine-tuning LLM backbone (learning rate 1e-6) since this is the most surprising result in our paper. Since LoRA fine-tuning or fine-tuning other parts (vision encoder or projector) is more regularized, doing validation study on the simplest fine-tuning LLM backbone is the most convincible choice.

| Config | Value |
|---|---|
| Optimizer | AdamW |
| Batch Size | 16 |
| Learning Rate Schedule | cosine decay |
| Warmup Ratio | 0.1 |
| Learning Rate | $1 \times 10^{-6}$ |
| Training Steps | 80000 |
| Freeze Vision Tower | True |
| Freeze Multi Modal Projector | True |
| Freeze Language Model | False |

(a) Configuration for ablation study across model size and model family, all the 3 models share the above hyper-parameters.

| Config | Value |
|---|---|
| Optimizer | AdamW |
| Batch Size | 16 |
| Learning Rate Schedule | cosine decay |
| Warmup Ratio | 0.1 |
| Learning Rate | $1 \times 10^{-6}$ |
| Training Steps | 20000 |
| Freeze Vision Tower | True |
| Freeze Multi Modal Projector | True |
| Freeze Language Model | False |

(b) Configuration for ablation study across rare datasets, all the 3 datasets share the above hyper-parameters.

| Config | Value |
|---|---|
| Optimizer | AdamW |
| Batch Size | 16 |
| Learning Rate Schedule | linear |
| Warmup Ratio | 0.1 |
| Learning Rate | $1 \times 10^{-6}$ |
| Training Steps | 2000 |
| Freeze Vision Tower | True |
| Freeze Multi Modal Projector | True |
| Freeze Language Model | False |

(c) Ablation study across dataset size, 2000 training steps corresponding to 2.5% dataset, the warmup steps is 2000*0.1=200. This configuration produce the results of 0.25% and 2.5%.

| Config | Value |
|---|---|
| Optimizer | AdamW |
| Batch Size | 16 |
| Learning Rate Schedule | linear |
| Warmup Ratio | 0.0025 |
| Learning Rate | $1 \times 10^{-6}$ |
| Training Steps | 80000 |
| Freeze Vision Tower | True |
| Freeze Multi Modal Projector | True |
| Freeze Language Model | False |

(d) Ablation study across dataset size, 80000 training steps corresponding to the 100% dataset, the warmup steps is 80000*0.0025=200. This configuration produce the results of 25% and 100%.

## C.3 Training Hyper-parameters for Figure 4

In this part, we still use full fine-tuning LLM backbone (learning rate 1e-6) as the default setting for the same reason with Appendix C.2. For hybriding different datasets, we use a fixed hybriding ratio of 0.5. The datasets will be oversampling if all the samples has been used at least once.

| Config | Value |
|---|---|
| Optimizer | AdamW |
| Batch Size | 16 |
| Learning Rate Schedule | cosine decay |
| Warmup Ratio | 0.1 |
| Learning Rate | $1 \times 10^{-6}$ |
| Training Steps | 80000 |
| Freeze Vision Tower | True |
| Freeze Multi Modal Projector | True |
| Freeze Language Model | False |

(a) Configuration for ablation study across hybriding different datasets and different hybrid ratio, all experiments share the above hyper-parameters.

## C.4 TRAINING HYPER-PARAMETERS FOR TABLE 3

We follow the configuration from MLLM-CL(Zhao et al., 2025) to achieve a fair comparison with their results.

| Config | Value |
| --- | --- |
| Optimizer | AdamW |
| Batch Size | 64 |
| Learning Rate Schedule | cosine decay |
| Warmup Ratio | 0.1 |
| Learning Rate | $8 \times 10^{-5}$ |
| Epoch for RS | 1 |
| Epoch for Med | 3 |
| Epoch for AD | 1 |
| Epoch for Sci | 2 |
| Epoch for Fin | 1 |
| LoRA rank | 8 |

(a) Hyperparameters of **IncLoRA** in MLLM-CL Benchmark *w/o replay buffer*.

| Config | Value |
| --- | --- |
| Optimizer | AdamW |
| Batch Size | 64 |
| Learning Rate Schedule | cosine decay |
| Warmup Ratio | 0.1 |
| Learning Rate | $8 \times 10^{-5}$ |
| Epoch for RS | 1 |
| Epoch for Med | 3 |
| Epoch for AD | 1 |
| Epoch for Sci | 2 |
| Epoch for Fin | 1 |
| LoRA rank | 16 |

(b) Hyperparameters of **IncLoRA** in MLLM-CL Benchmark *w/ replay buffer*.

| Config | Value |
| --- | --- |
| Optimizer | AdamW |
| Batch Size | 16 |
| Learning Rate Schedule | cosine decay |
| Warmup Ratio | 0.1 |
| Learning Rate | $1 \times 10^{-6}$ |
| Epoch for RS | 1 |
| Epoch for Med | 3 |
| Epoch for AD | 1 |
| Epoch for Sci | 2 |
| Epoch for Fin | 1 |

(c) Hyperparameters of **SeqFull** in MLLM-CL Benchmark *w/o replay buffer*.

| Config | Value |
| --- | --- |
| Optimizer | AdamW |
| Batch Size | 16 |
| Learning Rate Schedule | cosine decay |
| Warmup Ratio | 0.1 |
| Learning Rate | $1 \times 10^{-6}$ |
| Epoch for RS | 1 |
| Epoch for Med | 3 |
| Epoch for AD | 1 |
| Epoch for Sci | 2 |
| Epoch for Fin | 1 |

(d) Hyperparameters of **SeqFull** in MLLM-CL Benchmark *w/ replay buffer*.

## C.5 REPLAY BUFFER IMPLEMENTATION

We exactly follow the setting in **MLLM-CL** (Zhao et al., 2025), specifically, for each task of **RS, Med, AD, Sci, Fin**, we collect a replay data buffer of size 20 samples. Then, for every downstream sequential fine-tuning, we directly hybrid all the replay data of previous tasks into the current training data. No over-sampling mechanism is adapted.

# D EVALUATION PROTOCOLS

## D.1 PROMPT TEMPLATES

**Qwen2.5-VL.** We use the default `LLaMA-Factory` prompt, which is also the official prompt from `Qwen2.5-VL` repository.

> **System Prompt**
>
> user
> You are a helpful assistant. {*User's prompt*}
> assistant
> {*Model's response*}

**LLaVA-1.5.** We use the default `LLaMA-Factory` prompt, which is also the official prompt from `LLaVA-1.5` repository.

---

**System Prompt**

A chat between a curious user and an artificial intelligence assistant. The assistant gives helpful, detailed, and polite answers to the user's questions.
USER: {*User's prompt*}
ASSISTANT: {*Model's response*}

---

## D.2 EVALUATION OF 2X2 EVALUATION MATRIX

**Result Matcher.** We use a `result_matcher.py` file to evaluate the answer accuracy of predictions. All the questions in this part are multiple-choice questions and the answer is a single letter. All predictions are stored in a json file f, each `entry` has a `predict` key containing the model's output to the question and a `label` key containing a single letter as the ground truth. The logic is as follows:

```python
1  correct_predictions = 0
2  total_predictions = len(f)
3  for entry in f:
4      predict = str(entry['predict']).strip()
5      label = str(entry['label']).strip()
6
7      if ":" in predict:
8          predict = predict.split(":")[-1].strip()
9
10     predict = predict.upper()
11     label = label.upper()
12
13     if predict == label or predict.startswith(f"{label}."):
14         correct_predictions += 1
15
16  accuracy = correct_predictions / total_predictions
```

Listing 1: Pseudo code snippet for `result_matcher.py`.

This above script is adapted for evaluations curated from **ImageNet, Flowers 102, Caltech 101, Stanford Cars, ImageWikiQA**.

**VLMEvalKit.** For evaluation of **MMMU** and **VMCBench**, we directly use the code in VLMEvalKit (Duan et al., 2024) to get the results.

## D.3 EVALUATION OF RARE DATASETS

Since the questions we curated from **BSCCM** and **PitVis** are all multiple-choice questions, we use the same protocols as Appendix D.2, adapting the **Result Matcher** code in Listing 1.

## D.4 EVALUATION OF MLLM-CL

**Last and Average.** *Last* is the accuracy of all seen tasks after learning the last task. *Average* is the average accuracy of each task during the training process, *i.e.*, $Average = \sum_{i=1}^{t} acc_i$, where $t$ is the task that the model is learning, $acc_i$ is the accuracy of the $i$-th previous learned task.

**Result Matching.** For turning the generation result, we directly adapt the script from MLLM-CL to ensure the fair comparison. The only change is in the **Sci** script. The original script use the image storage path to distinguish different kind of types of questions, we find that this is detecting the multiple-choice question with one single choice letter as the ground truth. Thus, we replace the judge condition of `image.split('/')[-1].split('_')[0]=="AI2D"` or `image.split('/')[-1].split('_')[0]=="TQA"` or `image.split('/')[-1].split('_')[0]`

```
=="VQA"orimage.split('/')[-1].split('_')[0]=="SciVerse" with len(gt)
== 1.
```

**Evaluation code snippet for evaluating RS and AD.**   All the namings follows Appendix D.2.

```python
1  right = 0
2  total = len(f)
3  for entry in f:
4      ground_truth = entry['label']
5      if 'Unanswerable' in entry['predict'] :
6          continue
7
8      pred: str = entry['predict'].lower()
9      gt: str =  ground_truth.lower()
10
11     score = 0
12     if ' ' in gt:
13         if gt in pred:
14             right += 1
15     else:
16         gt = gt.replace('.', '')
17         if ' ' in pred:
18             if (' '+gt) in pred or (gt+' ') in pred or (gt+'.') in pred
   or (gt+',') in pred:
19                 right += 1
20         else:
21             if gt in pred:
22                 right += 1
23
24 accuracy = right / total
```
Listing 2: Pseudo code snippet for evaluating RS and AD.

**Evaluation code snippet for evaluating Med.**   All the namings follows Appendix D.2.

```python
1  right = 0
2  total = len(f)
3  for entry in f:
4      ground_truth = entry['label'].lower()
5      pred = entry['predict'].lower()
6      if 'Unanswerable' in entry['predict'] :
7          continue
8
9      if ground_truth in pred:
10         right += 1
11
12 accuracy = right / total
```
Listing 3: Pseudo code snippet for evaluating Med.

**Evaluation code snippet for evaluating Sci.**   All the namings follows Appendix D.2, the `prompt` key containing the question description.

```python
1  right = 0
2  total = len(f)
3  for entry in f:
4      ground_truth = entry['label'].strip()
5      problem = entry['prompt']
6
7      pred: str = entry['predict'].strip().lower().replace('.', '').replace
   (',', '').replace('neither', 'no')
8      gt: str =  ground_truth.strip().lower().replace('.', '').replace(',',
    '').replace('neither', 'no')
9
```

```
10      if len(gt) == 1:
11          if gt == pred:
12              right += 1
13      else:
14          if 'Which states' in problem:
15              gt_list = gt.split(',')
16              len_gt = len(gt_list)
17              pred_map_list = pred.split(',')
18
19              count = 0
20              for gt in gt_list:
21                  if gt in pred_map_list:
22                      count += 1
23              right += count / len_gt
24          elif gt in pred or pred in gt:
25              right += 1
26
27  accuracy = right / total
```

Listing 4: Pseudo code snippet for evaluating Sci.

**Evaluation code snippet for evaluating Fin.**    All the namings follows Appendix D.2.

```
1  right = 0
2  total = len(f)
3  for entry in f:
4      ground_truth = entry['label']
5
6      pred: str = entry['predict'].lower().replace(' ', '').replace('.', ''
       )
7      gt: str =  ground_truth.lower()
8      score = 0
9      if gt == pred:
10         right += 1
11
12 accuracy = right / total
```

Listing 5: Pseudo code snippet for evaluating Fin.

# E    FINE-TUNING ON PATH VQA

Experiment for full fine-tuning `Qwen-3B-Instruct` for 80,000 steps on the PathVQA dataset, which is an open-ended pathology question answering dataset. Since the fine-tuning process is independent of the ImageNet dataset, we think the only reasonable evaluation is the $OOD^T$-$OOD^I$ case, and the results are as follows:

|  | MMMU-dev (%) | MMMU-val (%) | VMCBench (%) |
|---|---|---|---|
| Before Fine-tuning | 48.00% | 48.44% | 75.20% |
| After Fine-tuning | 48.67% | 45.00% | 74.20% |

