# OpenReview forum: "Fine-tuning VLMs Without Forgetting Is Easier Than You Think"
_ICLR.cc/2026/Conference — Submitted to ICLR 2026_

### Official Review · Reviewer_4KR7 · 2025-10-19

**Soundness:** 4
**Presentation:** 4
**Contribution:** 3
**Rating:** 6
**Confidence:** 4

**Summary:**

This paper provides a systematic empirical study of catastrophic forgetting in vision-language model (VLM) fine-tuning. Rather than introducing a new algorithm, the authors argue that simple fine-tuning practices—notably small learning rates and parameter-efficient updates (e.g., LoRA)—are sufficient to prevent most forms of forgetting. Using a carefully designed 2×2 evaluation matrix (in/out-of-distribution text × image), they show that VLMs retain generalization across domains when regularized appropriately.
A notable exception is observed for in-distribution (ID) images with out-of-distribution (OOD) text, where forgetting manifests as task-specific overfitting—the model learns to ignore new prompts while retaining prior visual associations. To address this, the authors propose a data-hybrid training strategy that mixes small amounts of diverse instruction data to prevent overfitting. Extending to a continual learning benchmark (MLLM-CL), the same simple recipe outperforms or rivals complex continual-learning baselines without auxiliary mechanisms.

**Strengths:**

- **Comprehensive and well-controlled experimental design.** The 2×2 ID/OOD evaluation setup is elegant and isolates different axes of forgetting (image vs. text). The authors’ inclusion of both small- and large-scale VLMs (Qwen2.5-VL, LLaVA) strengthens the generality of the conclusions
- **Somewhat surprising empirical finding.** The claim that catastrophic forgetting in VLMs is largely overstated is both impactful and counterintuitive. It provides a fresh perspective likely to influence future methodology in multimodal fine-tuning.
- **Clear diagnosis of a new failure mode.** Identifying the ID-image / OOD-text regime as the key vulnerability (task-specific overfitting rather than capacity loss) is insightful and backed by solid diagnostic experiments (e.g., the class-label distractor test, Figure 3).
- **Simplicity and reproducibility.** The study uses minimal hyperparameter tuning, accessible setups (LoRA, low learning rates), and open-sourced code/datasets. This all ensures the paper’s transparency and educational value.
- **Strong continual-learning results.** The finding that simple sequential fine-tuning can match or outperform complex replay-based methods is remarkable and challenges current assumptions in the field

**Weaknesses:**

- **Limited insightful discussion.** While the empirical evidence is strong, the paper lacks a deeper explanation why VLMs are inherently robust to forgetting. Some discussion on model overparameterization, modularization between modalities, or representational sparsity would help ground the claims.
- **Missing connections to uncertainty-aware or Bayesian fine-tuning.** The discussion of regularization could benefit from referencing prior works that explicitly model uncertainty during (continual) fine-tuning, such as VPT [1] or CLAP4CLIP [2], which perform uncertainty-aware adaptation. This connection would situate the authors’ findings in a broader context of uncertainty-aware regularization.
- **Limited coverage of downstream tasks beyond VQA.** The study focuses on ImageNet-derived VQA formulations. It would strengthen the generality claim to evaluate on multimodal reasoning tasks (e.g., captioning, retrieval, grounded QA) where linguistic diversity is higher.
- **Contributions framing.** The paper repeatedly emphasizes “no new method,” but this undersells the significance of the findings. Perhaps, a reframing toward “rethinking the design space of multimodal adaptation” could better reflect its contribution.

**References:**

[1] Derakhshani *et al.* "Variational prompt tuning improves generalization of vision-language foundation models." Me-FoMo workshop, ICLR 2023.

[2] Jha *et al.* "CLAP4CLIP: Continual Learning with Probabilistic Finetuning for Vision-Language Models." NeurIPS 2024.

**Questions:**

- How does the proposed recipe interact with adapter stacking or mixture-of-LoRA methods (e.g., MR-LoRA, O-LoRA)? Would the same stability hold when multiple tasks are blended dynamically?
- In continual learning, how do your models behave under domain interference (e.g., conflicting visual distributions)? Would the same low learning rate recipe suffice without task identifiers?
- Can you quantify the epistemic uncertainty or representation drift across fine-tuning steps to empirically validate that “forgetting is not capacity-limited”? (This could also help connect well with the VPT/CLAP4CLIP uncertainty-aware continual learning literature.)
- Does the data-hybrid strategy work if the additional text data is synthetic (e.g., generated by GPT-4V)? This could test whether linguistic diversity or semantic content is the key factor.

---

> ### Author Response · Authors · 2025-11-22
>
> We sincerely thank the reviewer for the constructive feedback. We address the main concerns below.
>
> ---
>
> **Weakness 1 – Limited Insightful Discussion**: While the empirical evidence is strong, the paper lacks a deeper explanation for why VLMs are inherently robust to forgetting.
>
> **Response to Weakness 1:** We thank the reviewer for highlighting this important direction for improvement. We believe that factors such as representational sparsity and over-parameterization may contribute to the phenomenon. However, because our paper documents a *contradictory trend* in modern generative MLLMs compared to traditional VLMs and unimodal LLMs, our primary goal is to carefully characterize and analyze the phenomenon itself, and to identify which aspects of MLLMs lead to this behavior. Developing a full mechanistic explanation would be an excellent avenue for future work.
>
> ---
>
> **Weakness 2 – Missing Connections**: Missing connections to uncertainty-aware or Bayesian fine-tuning approaches, such as VPT or CLAP4CLIP.
>
> **Response to Weakness 2:** We thank the reviewer for pointing us to these valuable related works. However, these methods are designed for *traditional embedding-based VLMs* (e.g., CLIP), whereas our focus is on *modern generative MLLMs* (e.g., GPT/LLaVA), which do not provide explicit class-level logits. As a result, the techniques proposed in those papers are not directly applicable to our formulation. Although we include a classification-style setting for ablation, we aim for our findings to extend beyond classification tasks—which is one reason we include additional fine-tuning tasks in the continual learning section.
>
> ---
>
> **Weakness 3 – Limited Coverage Beyond VQA**: The study focuses on ImageNet-derived VQA formulations. Evaluating on multimodal reasoning tasks would strengthen generality.
>
> **Response to Weakness 3:** We appreciate the reviewer for raising this point. The MLLM-CL benchmark already includes non-VQA–style tasks (RS, Med, AD), which provide broader coverage. Moreover, because we are the first to report the non-forgetting phenomenon in generative MLLMs, we aim to evaluate it as reliably as possible. VQA-style tasks offer the clearest and most objective evaluation protocol. In contrast, tasks like captioning typically rely on subjective or model-based evaluators (e.g., GPT/Gemini), which introduce additional uncertainty.
>
> ---
>
> **Weakness 4 – Contributions Framing**: The paper frequently emphasizes “no new method,” which may undersell the importance of the findings.
>
> **Response to Weakness 4:** We thank the reviewer for this suggestion. We will revise the framing—particularly in the title and introduction—to better reflect the significance of our empirical findings.
>
> ---
>
> **Question 1 – Interaction with Other Methods**: How does the proposed recipe interact with adapter stacking or mixture-of-LoRA methods (e.g., MR-LoRA, O-LoRA)? Would the same stability hold when multiple tasks are blended dynamically?
>
> **Response to Question 1:** These methods can be viewed as more complex variants of simple incremental LoRA. They either introduce a router to select among LoRAs rather than merging them, or incorporate additional regularization when training new LoRAs. Our framework can be seen as a simplified version of these approaches, and we expect similar stability properties to hold.
>
> ---
>
> **Question 2 – Domain Interference**: In continual learning, how do your models behave under domain interference (e.g., conflicting visual distributions)? Would a low learning rate still suffice without task identifiers?
>
> **Response to Question 2:** The reviewer’s scenario maps directly to our evaluation matrix:
>
> * “Conflicting visual domains” correspond to the **ID(^T)–OOD(^I)** case,
> * “Conflicting textual domains” correspond to **OOD(^T)–ID(^I)**.
>   Our experiments in these settings already reflect such domain interference considerations.
>
> ---
>
> **Question 3 – Synthetic Data**: Does the data-hybrid strategy still work if the additional text data is synthetic (e.g., GPT-4V–generated)? This could help determine whether linguistic diversity or semantic content is the key contributing factor.
>
> **Response to Question 3:** Yes. The primary dataset used for hybridization—LLaVA-665K—is already a GPT-generated multimodal instruction-following dataset, demonstrating that synthetic data is effective in this setting.
>
> ---
>
> We thank the reviewer again for the constructive feedback and hope our responses adequately address the concerns raised.

---

> ### Author Response · Authors · 2025-11-26
> **Following up on our response to Reviewer 4KR7**
>
> Dear Reviewer 4KR7,
>
> We thank you again for your time and constructive feedback on our paper.
>
> We wanted to follow up to ensure you have had a chance to review our rebuttal. We have carefully addressed your concerns. We would appreciate it if you could let us know if our response has sufficiently resolved your concerns or if there are any remaining questions we can clarify before the discussion period ends.

---

### Official Review · Reviewer_yCvb · 2025-10-21

**Soundness:** 2
**Presentation:** 2
**Contribution:** 2
**Rating:** 2
**Confidence:** 3

**Summary:**

The paper provides experimental results to support that with appropriate regularization (low learning rate, constraining trainable params), VLMs prevent forgetting. Extensions to continual learning are shown, and practical guidelines.

**Strengths:**

- Paper considers an interesting evaluation protocol.
- Base models considered are near state of the art.
- Results on continual learning seem good.

**Weaknesses:**

- Recent published works, see [1] and references therein, demonstrate somewhat contradictory results to this paper. Through extensive empirical analysis across four state-of-the-art vision-language models and five fine-tuning techniques, a strong linear relationship holds: tasks with greater in-domain performance improvements suffer from more pronounced out-of-domain degradation, with some parameter-efficient fine-tuning (PEFT) methods exhibiting severe forgetting. A predictive measure (IIMM) also has been derived to capture when a VLM is expected to suffer more or less from forgetting. Even figure 2 in the paper shows there is some degree of forgetting.

[1] L. Niss et al, "The Inter-Intra Modal Measure: A Predictive Lens on Fine-Tuning Outcomes in Vision-Language Models", ICCV 2025, https://arxiv.org/abs/2407.15731

- No theoretical justification for results.
- Training recipes considered are well known existing works.
- Limited novelty contributions. The takeaways are overstated.

**Questions:**

- How does the degree of forgetting on off-target tasks correlate with on-target accuracy?
- For fine tuning tasks that yield large improvements post-fine tuning, what is the forgetting accuracy on previous general tasks?

---

> ### Author Response · Authors · 2025-11-22
>
> We thank the reviewer for the constructive feedback. We address the main concerns below.
>
> ---
>
> **Weakness 1 – Contradictory Results in Previous Work**: Recent published works, and references therein, report results that appear contradictory to ours: tasks with greater in-domain performance improvements often exhibit more pronounced out-of-domain degradation.
>
> **Response to Weakness 1:** We appreciate the reviewer for pointing us to this contradictory work. However, the cited paper focuses on *traditional* VLMs—specifically CLIP-style embedding models. Our paper, by contrast, examines *modern* VLMs (or MLLMs), which are **generative** models. While we are aware of several papers whose findings differ from ours, we have carefully reviewed them and found that all contradictory results occur in **unimodal LLMs, traditional embedding-based VLMs, or smaller-scale models**. To the best of our knowledge, ours is the first study to investigate this phenomenon in **modern generative MLLMs**, and the behavior we observe is unique to this model family.
>
> ---
>
> **Other Weaknesses – Lack of Theory, Novel Method, or Contribution**
>
> **Response to Other Weaknesses:** We would like to reiterate that our paper is intentionally positioned as an **empirical study**. Our primary goal is to document and analyze a surprising empirical phenomenon, rather than propose a new theory or algorithm. We are happy to refine or expand the paper if the reviewer can suggest specific directions where additional clarity or supporting analysis would be beneficial.
>
> ---
>
> **Question 1 – Relationship Between OOD Forgetting and In-Domain Accuracy**: How does the degree of forgetting on off-target tasks correlate with improvements in on-target accuracy?
>
> **Response to Question 1:** Our experimental results show that in **modern generative MLLMs**, fine-tuning improves on-target performance while causing **negligible forgetting** on off-target tasks. This holds across multiple datasets and settings, as demonstrated throughout our experiments.
>
> ---
>
> **Question 2 – Forgetting After Large Fine-tuning Gains**: For fine-tuning tasks that yield large performance gains, what is the extent of forgetting on previous general tasks?
>
> **Response to Question 2:** This question is addressed by the **OOD(^T)–OOD(^I)** entry in our evaluation matrix. We observe that general tasks remain **unaffected** by fine-tuning on a specific target dataset. In fact, we further show that the OOD(^T)–OOD(^I) setting is **more robust** than the OOD(^T)–ID(^I) case, and we provide an explanation for this behavior in the main text.
>
> ---
>
> We thank the reviewer again for the constructive feedback and hope our responses adequately address the concerns raised.

---

> ### Author Response · Authors · 2025-11-26
> **Following up on our response to Reviewer yCvb**
>
> Dear Reviewer yCvb,
>
> We thank you again for your time and constructive feedback on our paper.
>
> We wanted to follow up to ensure you have had a chance to review our rebuttal. We have carefully addressed your concerns. We would appreciate it if you could let us know if our response has sufficiently resolved your concerns or if there are any remaining questions we can clarify before the discussion period ends.

---

### Official Review · Reviewer_W4xs · 2025-10-29

**Soundness:** 3
**Presentation:** 3
**Contribution:** 2
**Rating:** 4
**Confidence:** 3

**Summary:**

The paper investigates catastrophic forgetting in the fine-tuning of vision–language models (VLMs) and shows that it can be largely mitigated through simple adjustments such as using smaller learning rates or limiting the number of trainable parameters. The authors introduce a 2×2 evaluation framework that systematically tests model performance across ID/OOD image and text settings. Their results confirm that these fine-tuning strategies effectively prevent forgetting in most cases, with the only notable failure occurring in the OOD-text / ID-image quadrant. Further analysis attributes this phenomenon to task-specific overfitting, which the authors mitigate using a data-hybrid training strategy that mixes diverse instruction data.Finally, the authors extend the fine-tuning strategies to continual learning setups and demonstrate the methods demonstrate strong performance among prior arts.

**Strengths:**

* The paper's presentation is both smooth and adequate, with experimental setups and core findings laid out nicely.
* The paper provides strong and comprehensive empirical evidence showing that catastrophic forgetting in VLMs is not inevitable and can be effectively mitigated with simple fine-tuning strategies such as small learning rates or parameter-efficient updates.
* The identification and in-depth analysis of the failure mode involving in-distribution images and out-of-distribution text are particularly insightful and offer an interesting understanding of task-specific overfitting within multimodal fine-tuning.

**Weaknesses:**

* Although the study covers a wide range of models and datasets, its fine-tuning experiments are largely restricted to classification-style tasks, which limits the generality of the conclusions. Such tasks may be relatively easy for modern VLMs, potentially requiring only minimal parameter updates to reach high accuracy. As shown in Figure 2, the model already achieves strong in-distribution performance even before fine-tuning, suggesting that the observed “robustness” may partly reflect task simplicity rather than intrinsic resistance to forgetting.

* The core finding that smaller learning rates or restricted parameter updates reduce forgetting—is not conceptually novel. For example, in [1] (which the paper cites), the author notice that "forgetting increases as a shifted power law in the number of parameters
fine-tuned and the number of update steps." and in [2], the authors find that using LoRA can mitigate catastrophic forgetting or stop (if limiting the training budget) in a lot of settings including instruction-tuning setups. Apart from these, modulating learning rate to mitigate catastrophic forgetting is also not a new topic to the continual learning domain.

[1] Kalajdzievski; Scaling Laws for Forgetting When Fine-Tuning Large Language Models.
[2] Biderman et al; LoRA Learns Less and Forgets Less.

**Questions:**

It is somewhat surprising and counterintuitive that in the continual learning experiments, the proposed methods achieve better performance without a replay buffer than with one, whereas all other baselines experience a performance drop under the same condition. Could the authors clarify the underlying reason for this behavior?

---

> ### Author Response · Authors · 2025-11-22
>
> We sincerely thank the reviewer for the constructive feedback. We address the main concerns below.
>
> ---
>
> **Weakness 1 – Easy Task Concerns**: The selected tasks are largely restricted to classification-style tasks and may be relatively easy for modern VLMs, potentially requiring only minimal parameter updates to reach high accuracy.
>
> **Response to Weakness 1:** We thank the reviewer for this observation. Regarding the reliance on classification-style tasks, we acknowledge that our fine-tuning primarily uses the ImageNet-VQA classification dataset. However, for general VQA tasks, constructing an apples-to-apples evaluation matrix is inherently difficult, as defining *“questions with the same visual input but different textual input”* is non-trivial. Our continual learning experiments already go beyond classification-style tasks.
>
> Additionally, we include an experiment where we fully fine-tune Qwen-3B-Instruct for 80,000 steps on the PathVQA dataset, which consists of open-ended pathology questions. Since this fine-tuning process is entirely independent of ImageNet, we consider the OOD(^T)–OOD(^I) setting to be the only reasonable evaluation. The results are shown below:
>
> |                    | MMMU-dev (%) | MMMU-val (%) | VMCBench (%) |
> | :----------------- | :----------: | :----------: | :----------: |
> | Before Fine-tuning |     48.00    |     48.44    |     75.20    |
> | After Fine-tuning  |     48.67    |     45.00    |     74.20    |
>
> Regarding the concern that the tasks may be too simple, we would like to emphasize that *simplicity does not preclude forgetting*. As shown in Figure 3, even fine-tuning on ImageNet-VQA—a relatively simple classification-style dataset—can still induce noticeable forgetting.
>
> ---
>
> **Weakness 2 – Observation Is Not Novel**: The core finding—that smaller learning rates or restricted parameter updates reduce forgetting—is not conceptually novel.
>
> **Response to Weakness 2:** We thank the reviewer and agree that *modulating the learning rate to mitigate catastrophic forgetting is a well-established idea* in the continual learning literature. However, prior work consistently reports a trend of *learning less and forgetting less* [1,2], whereas our results reveal a different and, to our knowledge, underexplored phenomenon: *learning the same amount without forgetting*. Our intention is to raise the question: **Are complex methods truly necessary for mitigating forgetting in MLLMs?** and to offer insight into **why MLLMs do not exhibit forgetting in the same way as LLMs**.
>
> [1] Kalajdzievski, *Scaling Laws for Forgetting When Fine-Tuning Large Language Models*.
> [2] Biderman et al., *LoRA Learns Less and Forgets Less*.
>
> ---
>
> **Question 1 – Counterintuitive Continual Learning Performance**: The proposed methods achieve better performance without a replay buffer than with one, whereas all other baselines experience a performance drop under the same condition.
>
> **Response to Question 1:** We thank the reviewer for noting this discrepancy. After re-examining the MLLM-CL paper, we believe the difference arises from the use of system prompts. In MLLM-CL, the implementation employs task-specific system prompts, whereas we use a single general prompt for all tasks (as described in Appendix D). Because of this, our model may experience degradation when training datasets are mixed—particularly when training is short and the model has not saturated—exactly the scenario in the MLLM-CL benchmark. From a benchmarking perspective, task-specific prompts naturally benefit performance. Thus, despite this discrepancy, the comparison on MLLM-CL still strongly supports the strength and simplicity of our proposed methods.
>
> ---
>
> We thank the reviewer again for the constructive feedback and hope our responses adequately address the concerns raised.

---

> ### Author Response · Authors · 2025-11-26
> **Following up on our response to Reviewer W4xs**
>
> Dear Reviewer W4xs,
>
> We thank you again for your time and constructive feedback on our paper.
>
> We wanted to follow up to ensure you have had a chance to review our rebuttal. We have carefully addressed your concerns. We would appreciate it if you could let us know if our response has sufficiently resolved your concerns or if there are any remaining questions we can clarify before the discussion period ends.

---

### Official Review · Reviewer_JeX7 · 2025-11-03

**Soundness:** 2
**Presentation:** 2
**Contribution:** 2
**Rating:** 4
**Confidence:** 5

**Summary:**

This paper argues that catastrophic forgetting in VLM fine-tuning has been overstated. Using a 2×2 evaluation framework (ID/OOD images × ID/OOD text), the authors show that simple methods, such as low learning rates (1e-6) or LoRA, can effectively preserve zero-shot capabilities while fine-tuning on ImageNet-VQA. They identify one failure mode: ID images with OOD text cause "task-specific overfitting" where models memorize templates instead of following instructions, which can be fixed by mixing diverse training data. In continual learning experiments, their simple approaches match or beat specialized methods, especially without replay buffers.

**Strengths:**

- **The 2×2 evaluation framework is genuinely useful.** Separately varying image and text distributions provides much finer-grained insights than typical "ID vs OOD" evaluations.
- **The systematic ablations are thorough and well-designed.** Table 1 shows that most regularization strategies fall within a ±3pp margin, which is exactly the kind of detailed analysis needed to support the claims. They also test across different model sizes (3B, 7B), architectures (Qwen2.5-VL, LLaVA), and rare domains (microscopy, surgery).
- **The practical value is real.** If the findings hold more generally, this could save researchers and practitioners a lot of unnecessary complexity. The message "just use a low learning rate" is actionable in a way that "implement this 10-component architecture" is not.

**Weaknesses:**

1.  **Limitations of classification tasks**: The paper's central conclusions (Findings I-III) are primarily validated on a single-task, classification-style VQA setup (ImageNet-VQA). While the continual learning experiments in Section 5 introduce some diversity, they do not sufficiently demonstrate that the proposed simple recipe generalizes to *single-task* fine-tuning on other complex, non-classification tasks (e.g., captioning, reasoning). This casts doubt on the universality of the main claim.

2.  **Conclusion can't be verified**: The attribution of the OODᵀ–IDᴵ failure to "task-specific overfitting" in the language module, while supported by a distractor experiment, remains somewhat indirect. The analysis does not fully rule out co-occurring factors, such as a potential degradation or narrowing of the visual representations for non-classification features. A more direct analysis (e.g., probing visual features or conducting ablations with a frozen vision encoder) would solidify this point.

3.  **Lack of Mechanistic Explanation**: The paper provides a strong empirical study but offers little insight into the *underlying reasons* for VLMs' apparent robustness compared to LLMs. Is it the inherent stability of the pre-trained vision encoder, the regularizing effect of the projector, or the multi-modal nature of the tasks themselves?

4.  **Insufficient Discussion of Related Work**
    The paper lacks a thorough discussion of several contemporary works that directly address catastrophic forgetting in multimodal models. For instance:
    - **"Model Tailor: Mitigating Catastrophic Forgetting in Multi-modal Large Language Models"** (ICML 2024)
    - **"LoRASculpt: Sculpting LoRA for Harmonizing General and Specialized Knowledge in Multimodal Large Language Models"** (CVPR 2025)
    - **"Locate-then-Merge: Neuron-Level Parameter Fusion for Mitigating Catastrophic Forgetting in Multimodal LLMs"** (Arxiv, 2025)

**Questions:**

See above.

---

> ### Author Response · Authors · 2025-11-22
>
> We sincerely thank the reviewer for the constructive feedback. We address the main concerns below.
>
> ---
>
> **Weakness 1 – Limitations of classification tasks:** The paper’s central conclusions are primarily validated on a classification-style VQA setup, which does not sufficiently demonstrate generalization to other complex, non-classification tasks.
>
> **Response to Weakness 1:** We thank the reviewer for highlighting the generalization concern. For evaluation, we design a 2×2 evaluation matrix that varies both visual and textual inputs. The textual inputs cover diverse question types beyond classification-style queries (ImageWikiQA — knowledge-based answering, MMMU — reasoning, VMCBench — vision-centric questions). On the fine-tuning side, we acknowledge that our primary experiments rely on the ImageNet-VQA classification dataset. However, for general VQA tasks, constructing an apples-to-apples evaluation matrix is inherently difficult since defining “questions with the same visual input but different textual input” is non-trivial.
>
> In addition to the continual learning experiments, we include an additional experiment where we fully fine-tune Qwen-3B-Instruct for 80,000 steps on the PathVQA dataset, which contains open-ended pathology questions. Because this fine-tuning process is independent of the ImageNet data, the only meaningful evaluation point is the OOD–OOD case. The results are:
>
> |                    | MMMU-dev (%) | MMMU-val (%) | VMCBench (%) |
> | :----------------- | :----------: | :----------: | :----------: |
> | Before Fine-tuning |     48.00    |     48.44    |     75.20    |
> | After Fine-tuning  |     48.67    |     45.00    |     74.20    |
>
> ---
>
> **Weakness 2 – Conclusion cannot be verified:** The analysis does not fully rule out co-occurring factors. More direct analysis (e.g., probing visual features or ablations with a frozen vision encoder) would strengthen the claim.
>
> **Response to Weakness 2:** We thank the reviewer for this suggestion. As stated in Appendix C, our fine-tuning recipe for analyzing task-specific overfitting already involves training with a frozen encoder. We apologize for any confusion arising from our use of the term “full fine-tuning,” which may have been misinterpreted. Since the phrase “LLM backbone fine-tuning” is not commonly used and could appear redundant, we refer to this configuration simply as “full fine-tuning,” in contrast to the other recipes considered.
>
> ---
>
> **Weakness 3 – Lack of mechanistic explanation:** The paper presents strong empirical results but provides limited insight into why VLMs appear more robust than LLMs.
>
> **Response to Weakness 3:** We thank the reviewer for encouraging a deeper discussion. We believe the natural robustness of MLLMs compared to LLMs stems from the high-dimensional and diverse nature of image inputs. As shown in *Finding 5*, forgetting emerges similarly to LLMs when the image input distribution is held fixed. Image tokens effectively “isolate” representations and help prevent the model from conflating unrelated knowledge. We view this as analogous to vocabulary learning: memorization aided by context, imagery, or pronunciation is typically more effective—and less prone to confusion—than rote memorization of a dictionary.
>
> ---
>
> **Weakness 4 – Insufficient discussion of related work:** The paper lacks a thorough discussion of contemporary works directly addressing catastrophic forgetting in multimodal models.
>
> **Response to Weakness 4:** We appreciate the reviewer for pointing out these relevant works. We will add a dedicated paragraph discussing them in the Related Work section. However, we emphasize that our focus differs from prior methods-oriented research: rather than proposing techniques to *mitigate* forgetting, we aim to investigate the more fundamental question of *whether forgetting occurs*. Due to space limitations, we do not include method-oriented papers as a central component of our discussion.
>
> ---
>
> We again thank the reviewer for the constructive feedback and hope that our responses adequately address these concerns.

---

> ### Author Response · Authors · 2025-11-26
> **Following up on our response to Reviewer JeX7**
>
> Dear Reviewer JeX7,
>
> We thank you again for your time and constructive feedback on our paper.
>
> We wanted to follow up to ensure you have had a chance to review our rebuttal. We have carefully addressed your concerns. We would appreciate it if you could let us know if our response has sufficiently resolved your concerns or if there are any remaining questions we can clarify before the discussion period ends.

---

### Meta-Review · Area_Chair_N1NG · 2025-12-24

**Summary:**

This paper received scores of 4,4,2,6. Concerns include issues with the approach and claims, questions around interpretability, missing related work, novelty, clarity, contradictory results, and limited discussion.  The rebuttal mostly addressed the concerns regarding clarity, missing related work, and contribution framing, but did not adequately address the other concerns, as detailed below.

**Reviewer Concerns:**

Reviewer JeX7's concern regarding generalization beyond classification tasks is only partially addressed by the rebuttal. In particular, demonstrating single-task fine-tuning on multiple diverse non-classification tasks is missing from the paper and rebuttal.  Similarly, reviewer W4xs's concern about easy classification task is only partially addressed by the rebuttal.  It does not adequately address the concern that robustness may partly be an artifact of task easiness.

Reviewer yCvb’s concern regarding contradictory results with prior work is not adequately addressed. In particular, there is no response regarding IIMM or Figure 2.  Reviewer 4KR7's concern around limited coverage beyond VQA is only partially addressed; in particular, the generality of the claims is not fully supported.

**Reviewer Scores:**

Based on the rebuttal, the reviewers may have kept their scores as-is.

---

### Decision · Program_Chairs · 2026-01-26

Reject